# Determining the Relationship of Meteorological Factors and Severe Pediatric Respiratory Syncytial Virus (RSV) Infection in Central Peninsular Malaysia

**DOI:** 10.3390/ijerph20031848

**Published:** 2023-01-19

**Authors:** Chee Mun Chan, Asrul Abdul Wahab, Adli Ali

**Affiliations:** 1Department of Pediatric, Faculty of Medicine, Universiti Kebangsaan Malaysia, Jalan Yaacob Latif, Kuala Lumpur 56000, Malaysia; 2Department of Microbiology, Faculty of Medicine, Universiti Kebangsaan Malaysia, Jalan Yaacob Latif, Kuala Lumpur 56000, Malaysia

**Keywords:** RSV infection, children, rainfall, humidity, climate

## Abstract

Respiratory syncytial virus (RSV) is the most common pathogen causing viral respiratory tract infections among younger children worldwide. The influence of meteorological factors on RSV seasonal activity is well-established for temperate countries; however, in subtropical countries such as Malaysia, relatively stable temperate climates do not clearly support this trend, and the available data are contradictory. Better understanding of meteorological factors and seasonality of RSV will allow effective strategic health management relating to RSV infection, particularly immunoprophylaxis of high-risk infants with palivizumab. Retrospectively, from 2017 to 2021, we examined the association between various meteorological factors (rainfall, rainy days, temperature, and relative humidity) and the incidence of RSV in children aged less than 12 years in Kuala Lumpur, Malaysia. RSV activity peaked in two periods (July to August and October to December), which was significantly correlated with the lowest rainfall (*p* < 0.007) and number of rainy days (*p* < 0.005). RSV prevalence was also positively associated with temperature (*p* < 0.006) and inversely associated with relative humidity (*p* < 0.006). Based on our findings, we recommend that immunoprophylaxis with palivizumab be administered in children aged less than 2 years where transmission of RSV is postulated to be the highest after the end of two monsoon seasons.

## 1. Introduction

Acute respiratory tract infections (ARTIs) are a global contributor to rising morbidity and mortality among children less than five years of age [1]. According to the worldwide data collected in 2017, 650,000 children from this age group succumb to death from lower respiratory tract infections (LRTIs) every year, especially those living in low-income countries or exposed to low socioeconomic environments [2]. Although there are many pathogens causing ARTIs such as bacteria, viruses, and fungi, many studies cited that most infections in this younger age group are caused by viruses, commonly the respiratory syncytial virus (RSV) [3]. According to the latest report by the World Health Organization (WHO) published in 2019, the global burden of RSV-associated ARTIs is estimated at 33 million annually, with more than 3 million hospitalizations and 56,600 in-hospital deaths in children aged under 5 years. What is more worrying, infants under 6 months of age contributed to nearly half of the cases, with 1.4 million hospitalizations and 27,300 in-hospital deaths annually [4]. Besides age, underlying conditions such as prematurity, lung disease, or congenital heart disease can also predispose children to RSV infection [5,6].

Respiratory viruses, such as RSV, are usually transmitted through inhalation or direct contact with secretions or aerosols [7]. With its lipid-enveloped property, the transmission and viral activity are often associated with geographic and meteorological factors [7,8]. In temperate countries with distinct seasonality, many studies have reported the association of RSV activity with temperature, humidity, and rainfall. A retrospective study in Spain demonstrated a seasonal pattern of RSV admission from November to March, which coincided with weather transition from fall to winter. Similarly to many other studies, the authors found an inverse relationship between mean temperature and activity of RSV. It was explained that RSV inactivation required a longer time when temperature decreased but was shortened when temperature rose [9]. This trend was also observed in a five-year-long study by Bing Xu et al. where they reported a clear seasonal variation in RSV-associated admission with peaks in winter months, from November to March. Cold weather was postulated to lower the physical protection barrier of the human body by decreasing the secretion function of the respiratory mucous membrane and weakening the body’s immune system and function, thus causing infection [3]. Apart from outdoor meteorological variables, it is also important to consider human behavior and actions in response to change of weather. During summer with a higher relative humidity, indoor crowding in air-conditioned environments has commonly been speculated to be a key reason for indoor transmission, causing a rise in RSV cases [3,8]. However, as indoor data are scarce, outdoor climate factors have been used as surrogate markers to indirectly explain the influence of indoor factors.

In recent years, emerging evidence suggests that tropical regions, defined as areas with predictable dry and wet (rainy) meteorological seasons throughout the year, may not have the same seasonal peaks or outbreaks of RSV infection [1]. Generally, RSV infections are observed year-round, with peak activity occurring during rainy seasons [1]. Malaysia is a tropical country that experiences warm and humid climate conditions throughout the year. Sitting in the monsoon belt of Southeast Asia, it is influenced by the Southwest monsoon (May to August) and the Northeast monsoon (November to February), with the estimated total rainfalls of 2000 mm to 3000 mm annually [10]. A local study in Malaysia by Khor et al. in 2008 cited that RSV showed prominent seasonality, with peak activity in the rainy season (September to December) [7]. Neighboring countries, e.g., Thailand, also supported this finding, and the authors hypothesized that temperature and relative humidity are inversely associated with disease activity, supporting the preconceptual ideas in tropical countries. Unfortunately, despite regional proximity and similar climates, the epidemiological pattern of RSV infection in Southeast Asia is inconsistent and exhibits subtle differences, suggesting that climate factors alone do not dictate RSV seasonality [11]. Subsequently, in 2013, another local epidemiological study by Rahman et al. reported an opposite result, whereby the peak of RSV activity was observed from May to June [12]. This result is further supported by another 5-year-long study in Kuala Lumpur, according whereto peak infection occurred from July to September [13]. This was also true for Singapore and Lombok, Indonesia, where the peak of RSV cases also did not coincide with their rainiest season, with most RSV cases recorded from March to August [14,15]. Thus, the timing of epidemics of RSV is indeed diverse in tropics and more data are needed to establish the RSV activity.

Despite the commonality and disease burden among younger children known from numerous worldwide reports, there are few epidemiological studies on RSV infection in Malaysia. To our best knowledge, the latest study to evaluate the association of climate factors and RSV activity was carried out in 2008 [7]. The amount of up-to-date data is insufficient for us to establish the pattern of activity of RSV, especially with increasing global warming, attributing to the country’s active development throughout the years. The Malaysian Meteorological Department reported that the highest temperature in Malaysia was observed in 2016, attributed to a Super El Niño event [10]. To bridge the knowledge gap, a clearer understanding of the relationship between seasonality of viruses and meteorological factors is pivotal to successful implementation of prevention and control programs. Hence, this retrospective study aimed to describe the seasonality and association of RSV viruses with meteorological factors based on the data extracted over the last five years from a tertiary teaching hospital in Kuala Lumpur, a central region in Peninsular Malaysia.

## 2. Materials and Methods

### 2.1. Study Design and Sample Size

This was a retrospective study based at Hospital Canselor Tuanku Muhriz (HCTM), Kuala Lumpur. We obtained previously collated data for a 5-year period from 1 January 2017 to 31 December 2021. During the 5-year period, all patients (aged 0 to 12 years) with NPA analysis were included in this study, and further demographic information including date of birth, race, test date, and age at time of test was extracted from the patient databases. Children who presented with mild respiratory symptoms, did not require NIV, or with no NPA analysis were excluded from this study. Besides, we secondarily excluded repeated RSV samples obtained from the same patient if they were taken within 2 weeks. This is because they were considered to be within the same period of infection, hence only the earlier RSV sample would be taken for analysis. In addition, it is also worth highlighting that our study population was affected by the COVID-19 pandemic, which might have undermined the actual prevalence of the infection in 2020. The restriction of activities due to COVID-19 or the movement control order (MCO) was strictly implemented nationwide on 18 March 2020, prohibiting any mass gatherings or movements to limit the transmission of the disease [16]. During this period, only one individual per household was allowed to travel within a 10-km radius to purchase essential items. All activities, including education and work, were prohibited, restricting children and adults to remain at home all the time. The first phase of the strict MCO lasted about 2 months, followed by the conditional MCO for a month, which only enabled adults to work for the benefit of the country’s halted economy. With a gradual reduction of cases, the Malaysian government revised its plan to introduce the recovery MCO from 10 June until 31 December 2020. Most of the activities were resumed in phases, with mandatory enforcement of social distancing, vaccination, and personal protection for all Malaysian citizens. It is also worth highlighting that 5 months after their first implementation, all travel and gathering bans were eased on 29 August 2020, promoting domestic tourism for the growth of the country’s economy. In 2021, the MCO was lifted entirely, and Malaysians were able to resume their pre-COVID lifestyles. Nevertheless, most of the citizens still observe self-protection with facemasks while engaging in daily activities [16]. This study was approved by the UKM’s Ethics Committee (code FF-2021-452).

### 2.2. RSV Infection Detection

ARTIs are defined as the presence of cough and cold with respiratory symptoms such as fast breathing, tachypnoea above age limit, and/or chest in-drawing and danger signs such as being unable to feed, persistent vomiting, lethargy, or stridor [17]. Symptomatic children with moderate to severe respiratory distress symptoms requiring non-invasive ventilation (NIV) were admitted, and nasopharyngeal aspirate (NPA) was usually taken to detect various common respiratory viruses. Direct fluorescent antibody (DFA) method using D3 Ultra DFA Respiratory Virus Screening and Identification Kits (Diagnostic Hybrids, USA) were utilized for the identification of RSV, adenovirus, influenza A and B and parainfluenza 1, 2, and 3 viruses, with the sensitivity of 95.5% and the specificity of 98.3%. Briefly, the nasopharyngeal cells obtained from NPA were added to the DFA screening reagent to determine the presence of viral antigens using fluorescence microscopy. Once the stained cells showed positive apple-green fluorescence, the particular virus was further identified using individual virus-specific DFA reagents [17]. In our study, our primary interest was to detect RSV, a single-stranded negative-sense RNA enveloped virus which commonly causes viral bronchiolitis and pneumonia in infants and children. There are two major subtypes, A and B, where B is characterized as an asymptomatic strain that the majority of patients experience. Usually, the more severe clinical illness predominantly involves the subtype A strain, especially in most outbreaks [18]. Over the years, RSV has been commonly detected directly in cells from the nasopharyngeal epithelium by staining with immunofluorescent reagents, although it can also be isolated in certain cell cultures.

### 2.3. Meteorological Data

Monthly meteorological data were obtained from the Malaysian Meteorological Department, Ministry of Environment and Water, for the stipulated period. The distance between the meteorological station (3°06′07″ N, 101°38′42″ E) and HCTM is approximately fourteen kilometers. Among the meteorological parameters included were monthly rainfall, rainy days, temperature, and relative humidity.

### 2.4. Statistical Analysis

Data analysis was performed using Statistical Package for the Social Sciences (SPSS) version 26 (IBM, Chicago, IL, USA). Descriptive statistics were used to describe demographic data and expressed as frequencies and percentage for categorical variables while continuous variables were expressed as median with range. For better approximation, meteorological data throughout the 5 years were calculated as the means and aggregated into months for further analysis. The relationship of positive RSV cases with meteorological factors was plotted on time series graphs to visually illustrate the trend of infectivity. We also statistically confirmed the association between the two variables using Spearman’s rank correlation. Significant correlations of meteorological factors were later confirmed with bivariate logistic regression analysis. A two-sided *p*-value of less than 0.05 was considered statistically significant.

## 3. Results

### 3.1. Demographic Data of Severe RSV Infection in the Past 5-Year Period

Over the 5-year period (2017–2021), 2950 samples obtained from children admitted with symptoms of lower respiratory tract infections (LRTIs) were sent for respiratory syncytial virus (RSV) detection, of which 444 (15.1%) were positive for RSV. Nine repeated RSV samples over a period of 2 weeks were excluded, making the final sample size of 435 positive RSV cases (14.8%). Demographically, the median age of the positive cases was 1 year, and the majority of them were aged less than 2 years, with peak incidence among the children under 6 months (20.1%). We also reported that the numbers of RSV-positive cases were increasing from 2017 to 2019 (8.9–17.7%) and later experienced a slight dip in 2020 before peaking in 2021 (20.2%). To illustrate the effect of the COVID-19 pandemic which possibly explained the result in 2020, we stratified our study population into two cohorts: pre-COVID-19 and post-COVID as seen in Appendix A. Although RSV infection had the highest prevalence in children aged younger than 2 years, the post-COVID group (2020–2021) reported a higher incidence in older children aged between 6 months to 2 years compared to the pre-COVID group. We hypothesized that older children were easier to bring to healthcare facilities for appropriate diagnosis and treatment compared to infants, especially in view of the worrying COVID-19 pandemic. Nevertheless, with restricted movement explained in the discussion later, we agree that our findings may underreport the actual RSV prevalence in 2020 due to the COVID-19 pandemic. Overall, the majority of RSV cases involved Malay children (15.4%) while Indian and children of other ethnic origin were less affected. The demographic data of the RSV-positive and RSV-negative cases are summarized in Table 1.

### 3.2. The Relationship between RSV Infection and Amount of Rainfall; Number of Rainy Days

Within these five years, two distinct periods recorded notably higher amounts of rainfall and numbers of rainy days in Kuala Lumpur, Malaysia, as seen in Figure 1a,b. For a better phenomenon understanding of the monsoon season, Appendix A showed that the highest amounts of rainfall and numbers of rainy days were observed from October to December, ranging between 104 mm to 678 mm and 11 to 29 days, respectively. Although the wet season was also observed in the first monsoon season of the year from March to May, much less rainfall (72–598 mm) and fewer rainy days (9–23 days) were reported.

To depict a clearer picture of seasonality within a year, the number of RSV cases was stratified into months. RSV infections were present throughout the year and demonstrated pronounced seasonality, with rising infections from year-end (October to December) to January and mid-year (July to August). Visually, from Figure 1a,b, we observed that there was an inverse relationship between the number of cases and rainfall and rainy days. Therefore, we further analyzed the association of RSV with climate factors, as seen in Table 2. Spearman’s rank correlation and bivariate regression analysis showed significant indirect correlation of monthly RSV cases with (i) rainfall and (ii) rainy days. The regression model postulated that for an increase of one millimeter rainfall or one rainy day, the number of RSV cases significantly decreased by 0.001 and 0.004, respectively. This hypothesis was supported by our Appendix A to determine the individual effects of each monsoon season. Interestingly, we also reported significant correlations of these two factors with RSV seasonality in the first monsoon season (*p* < 0.001) since the first monsoon has a better variability in the amount of rainfall and the number of rainy days. Comparatively, this trend was not observed in the second monsoon season due to the relatively stable wet conditions throughout 3 months.

### 3.3. The Relationship between RSV Infection and Temperature; Relative Humidity

The temperature was relatively stable throughout the year, ranging between 26.7 °C and 29.4 °C, with the highest average temperature recorded in July (28.9 °C) and the coolest—in December (27.8 °C). Additionally, relative humidity tends to be the highest from November to December (75.8% to 77.1%), coinciding with the wettest season in Central Peninsular Malaysia. Interestingly, we noted that the number of cases appeared to rise with increasing temperature and was inversely associated with relative humidity, as seen in Figure 1c,d. This trend is supported statistically with a positive correlation between RSV infection and temperature where a 1 °C increase in temperature increases the number of cases by 0.03, as seen in Table 2. As relative humidity is inversely associated with rising temperature, we observed that a 1% drop of relative humidity was associated with a 0.005 increase in RSV cases. In addition, a separate analysis of the two monsoon seasons with different average temperature showed that the colder second monsoon season (28.3 °C) contributed to a significant increase in RSV transmission (*p* < 0.007). The drastic difference of temperature attributed to higher rainfall and rainy days throughout the whole monsoon season definitively showed the importance of different meteorological conditions that affect RSV transmission.

## 4. Discussion

According to a health report by the Malaysian Ministry of Health in 2020, disease in the respiratory system was ranked as one of the top three major causes of hospitalization, with 9.01% and 11.55% of admissions in public and private hospitals, respectively [19]. Similarly to many other studies, our result also highlighted that children younger than two years are more susceptible to RSV infection, and the positive detection rate decreased tremendously as the age of children increased [12,13]. This can be partly explained by the diminishing role of natural passive maternal immunity within months after birth, posing a higher risk to infants aged younger than six months. RSV infections are indeed a common infection among children, with nearly all children being infected with RSV at least once during the first two years of life, with about two-thirds being affected by the end of their first year [5]. Thus, immunological protection is eventually acquired through early infections or vaccination, resulting in a decline of positive cases among older children [5,7,13]. We also hypothesized that the rate of positive detection of RSV increased gradually every year, from 9% to 20.2% within the five years of study. Unfortunately, the restriction of activities during the COVID-19 pandemic, the implementation of the MCO, and mask wearing impacted the actual RSV prevalence and its association with meteorological factors [16]. We established two cohorts, namely the pre-COVID cohort (2017 to 2019) and the post-COVID cohort (2020–2021), to demonstrate the effect of COVID-19 on RSV prevalence. From our Appendix A, we noted significant correlations of RSV seasonality with all the four meteorological factors, similarly to our pre-conceptualized idea. However, our analysis of the second cohort showed no significant correlation with climate factors, although the trend was similar. The second cohort had approximately two times fewer positive RSV samples than the first cohort. During the rise of COVID-19 cases with various phases of the MCO, the cases reported in 2020 were reduced to 15.0%, which we believe to be underreported due to the limitation of access to healthcare services with the nationwide lockdown [16]. Upon lifting of the control order in 2021, the number of RSV cases peaked exponentially to 20.2% as children resumed their activities with peers. Thus, a larger sample size is required for future studies to comparatively demonstrate the effect of COVID-19 on RSV seasonality. Nevertheless, we agree with the findings from our local colleagues Low et al. who also reported an increase of 517.4% in RSV cases from 2015 to 2019 [13].

Pathogenically, a few studies demonstrated that seasonality of RSV is more pronounced, with a higher correlation after a period between two weeks to two months of weather exposure [6,11]; this is considering the infectious period of ten to fourteen days (including the incubation period and virus shedding) and the time taken for older children to transmit RSV to their younger siblings. We attempted to demonstrate a stronger association between RSV prevalence and meteorological factors by lagging forward the climate factors by one and two months, respectively [6]. Interestingly, our results were not significant for all the four factors after adjusting the meteorological variables (*p* > 0.05) as mentioned. This can be due to our expression of meteorological data on a monthly basis instead of the weekly one, which was not sufficiently discriminatory to estimate the duration of the incubation period after weather exposure. Nevertheless, we agree with some studies where the highest associations were observed with no time lag as theoretically it only takes ten to fourteen days for RSV activation [11].

Our data showed that RSV infections were detected year-round, with the most distinct seasonality of peak infections occurring mid-year (July to August) and at year end (November to January). This is supported by a few local and regional articles that were published earlier. Two epidemiological studies in Kuala Lumpur discovered that pronounced infection peaks were detected from June to September, while an earlier study showed RSV peaked at the end of the year [5,7,13]. As described earlier, there are two monsoon seasons in Malaysia, and interestingly, we found that our results of rainfall and rainy seasons coincided earlier than the reported fact of Southeast Asia (SEA) monsoon seasons. Referring to climate change due to rapid urbanization of the country, studies explained that it may be due to the urban heat island effect, which increased rainfall variability due to facilitated deep convection process [20]. Wang et al. demonstrated that extreme precipitation in Asian monsoons exhibited the highest sensitivity to global warming, with increased frequency of rainfall at much higher rates [20]. With that, we also agree with Tanita et al. in that rainy seasons in SEA monsoons tend to start earlier than the typical April or May, resulting in a slightly longer wet season [21]. With regard to RSV infection, this trend of infection was likely to be affected by meteorological factors as observation showed a peak after the rainy seasons. Generally, there are four main aspects to explain the relationship between meteorological factors and respiratory viruses including (i) virus’ survival, (ii) virus-to-host infection; (iii) virus transmission; and (iv) host’s immune response [3]. These meteorological factors could directly affect viral stability, viability, activity, and survival time [3]. In a study by Teck et al., a significant positive association between rainfall and RSV infection was reported, with 55% of the positive cases coinciding with heavy rainfall [5]. This finding was in agreement with an earlier study in Thailand which reported rainfall had the highest positive correlation to RSV infection. On the contrary, Khor et al. found that the number of rainy days was significantly associated with RSV cases, unlike rainfall [7,11]. They postulated that the absolute amount of rainfall did not reflect the consistency of rain throughout the entire month as it can be brief, intense showers or prolonged episodes of light rainfall. Interestingly, we also demonstrated a stronger association of rainy days with RSV cases than that of the amount of rainfall. However, our findings did not correlate with the previous studies, as we reported an inverse relationship between rainfall and rainy days and RSV cases. We showed that RSV seasonality peaked at the end of the two monsoon seasons, which were July to August and December to March, respectively. A local meteorological study by Tang et al. reported that the first monsoon season (May to August) featured a drier weather with less rainfall compared to the second one [22]. Thus, this could possibly explain the contrasting result as children were more exposed to the outer environment such as public events, activities, and gatherings in fair and bright weather, especially in conjunction to the annual mid-term school holidays which lasted about 2 weeks and 1 week in the middle of the months of June and August, respectively [23,24,25]. We postulate that less rainfall and fewer rainy days resulted in a higher participation of children in outdoor activities, thus increasing the risk of RSV transmission.

On the contrary, the second monsoon season has a higher amount of rainfall and more rainy days than the first. The relatively constant wet weather conditions throughout with rainy days for two-thirds of the month on average result in a colder outer environment. As a result, most of the children are kept warm indoors. We opposed the preconception that staying indoors during rainy days increases transmission of respiratory viruses as indoor variables are commonly cited as a subjective measurement to gauge their effect on the transmission [3,8]. Indoor environment includes a broad range of locations such as shopping malls and indoor gatherings, and these factors were never properly investigated in the available literature [8]. At the same time, towards the year end, in December, when school holidays begin and working parents start clearing annual leaves, wet outdoor conditions possibly promote children gathering in enclosed spaces such as shopping malls, thus increasing the transmission rate of RSV [24,25]. The festive season of Christmas and New Year with delightful decorations in December encourages children to visit shopping malls. Hence, the RSV infection peaks highest in December and January after the increased transmission rate among children.

The relationship between relative humidity and temperature is well-established. Relative humidity is defined as the percentage of water vapor in the air that changes when the temperature changes [26]. They are inversely associated because as air temperature increases, relative humidity decreases as the air can hold more water molecules [26]. Laboratory studies have successfully shown that the lipid-enveloped property of RSV allows better survival in cooler and humid environments [8]. The inverse relationship between temperature and RSV survival can be explained by its prolonged inactivation time when temperature is decreased. A study in Spain observed an increase in the activity of RSV when temperature dropped to below 9 °C. This was also similar to another study in China which demonstrated temperature had the greatest explanatory power affecting RSV infection [9]. With the inhalation of cold and dry air in cold weather, host defense mechanisms can also be altered by the cooling of the nasal cavity, impairing mucociliary activity and lowering the phagocytic activity of leukocytes, which may result in a better survival and viability of RSV [3]. Unfortunately, the effect of temperature on RSV cases in tropics was subtle compared to temperate countries, mainly due to the relatively stable and higher temperature with no drastic differences throughout the year [1]. We observed that higher temperature with reduced humidity increases the transmission of RSV cases. Air-conditioned facilities such as shopping malls and function halls have been mushrooming over the years, and many opt to stay indoors when the outside temperature increases [3,8,14]. This change of social behavior will indirectly contribute to increased transmission of viruses due to overcrowding of susceptible individuals indoors with poor ventilation [14]. On top of that, air conditioning also lowers the indoor temperature, making the environment more favorable for RSV survival using a similar concept as explained above. Chew et al. also reported the same trend, and they hypothesized that it was possibly contributed to by the increased number of visitors escaping from temperate climates to enjoy hotter and dry seasons in Singapore [27]. Thus, we conclude that rising temperature and reduction in humidity indirectly mobilize people to gather indoors and increase RSV transmission, contributing to the rise of cases.

Our study supports the recommendations to consider and update the local epidemiological factors affecting the rate of RSV transmission. This is important to determine the timing of administration of palivizumab to children to reduce the risk of moderate to severe RSV infection. Currently, there are neither specific treatments for RSV illness nor approved methods of active immunization [28]. It is primarily symptomatic and includes oral and parenteral fluid replacement to ensure adequate hydration, supplemental oxygen or mechanical ventilation in cases of moderate to severe respiratory illness, and a nasal decongestant [29]. Thus, the clinical standard for RSV prevention is passive immune-prophylaxis with palivizumab, a monoclonal antibody to the RSV F-protein targeted to reduce the fusion of the virus with cell membranes and avoid the formation of syncytia in the lungs by preventing cell-to-cell spread of the virus [29]. Studies have recommended that palivizumab be administered monthly by intramuscular injection during the RSV season in order to maintain therapeutic serum levels. However, there are discrepancies in the injection interval whereby some studies were concerned with breakthrough infections and adherence, hence it has been postulated to shorten the interval to 16–24 days instead of 30 days [27]. With limited epidemiological studies of the trend of infections pertaining to meteorological factors which are rather predictable yearly in Malaysia, there is no proper guideline developed for the optimal use of prophylaxis. More importantly, considering epidemiological variability between places and countries especially at the time of global warming, it is recommended that preventive strategies (palivizumab, but also nirsevimab) be outlined according to the latest local data of RSV seasonality and prevalence. As a result, this highlights the importance of our study to determine the seasonality of RSV infections which can bridge another research gap of optimal timing and interval duration of palivizumab therapy to effectively reduce the number of cases.

There are several limitations to consider in our study. Firstly, as a retrospective 5-year-long study, it was not possible to ensure the consistency of nasal swab testing for all the children presenting with symptoms of acute respiratory infections, although there were no changes to the usual practice recommendations made during this period. Additionally, our findings do not reflect the entire population and RSV seasonality in Malaysia as different climatic conditions may be experienced in other states of the country, particularly in those nearer to coastal areas. Hence, a larger dataset of reliable and more comprehensive epidemiological data should be compiled via regional collaboration of multistate sites with different climatic conditions. Thirdly, our RSV detection kit (D3 Ultra DFA Respiratory Virus Screening and Identification Kit) also has several limitations [18]. Among them are false negative results due to inappropriate specimen collection, storage, or handling and the effects of empirical antiviral therapy. Besides, the detection of viruses also varies greatly depending on the specimen’s quality and the handling technique as a negative result does not exclude the possibility of virus infections; thus, it requires consideration of other information such as epidemiological studies and clinical evaluation of the patient [18]. Fourthly, our meteorological data were recorded in months instead of in weeks, hence they may not effectively provide a reliable forecast of the onset and incubation period of RSV which are useful for public health authorities to prepare and respond to RSV epidemics. In addition, there are also many confounding factors which influence the association between meteorological factors and RSV activity. Factors such as indoor climate and human behavioral response were not proven definitively despite our hypothesis of their contribution to this relationship. We believe that the true mechanism of influence of meteorological factors on RSV might be very complex, thus further studies should consider the effect of indoor and outdoor climate on respiratory infections.

## 5. Conclusions

In conclusion, this is the latest Malaysian study to demonstrate RSV seasonality in a tropical country and its association with local meteorological variables. We report that RSV peaked twice in Malaysia, between July and August and from October to December, respectively. We believe that this study will help to improve the knowledge of virus transmission in different climatic environments, hence reducing the number of cases by optimally administering immunoprophylaxis to high-risk children. Thus, we suggest that prophylaxis with palivizumab should be considered in children aged younger than 2 years within these two periods which coincide with the lowest amount of rainfall, number of rainy days, and relative humidity. We hope to explore further the understanding of meteorological effects in guiding stakeholder policy planning through time-efficient public health interventions and facilitate the implementation of future vaccination and immunotherapy strategies.

## Figures and Tables

**Figure 1 ijerph-20-01848-f001:**
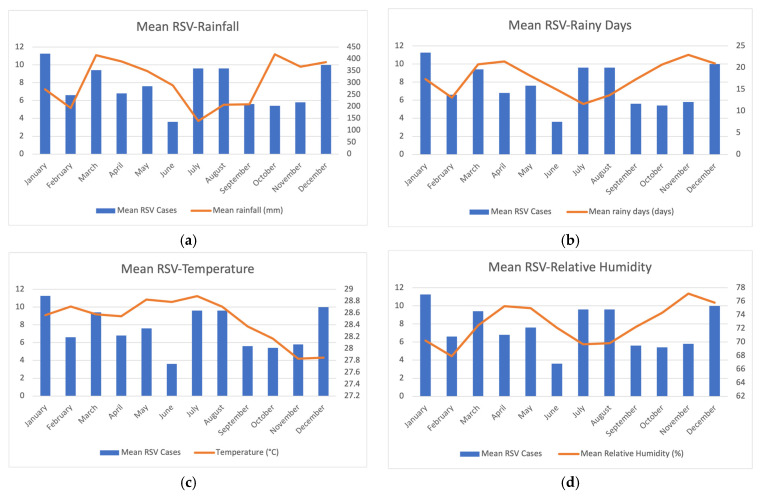
(**a**–**d**) Average monthly distribution of RSV-positive cases and various meteorological factors over the period of 5 years (2017–2021).

**Table 1 ijerph-20-01848-t001:** Characteristics of the study population.

Indicator (N = 2941)	RSV-Positive, n (%)	RSV-Negative, n (%)
Nasopharyngeal swab result	435 (14.8)	2506 (85.2)
Age (years)		
<6 months	117 (20.1)	466 (79.9)
6 months to 2 years	234 (17.9)	1067 (82.1)
2.1 years to 5 years	60 (7.4)	752 (92.6)
>6 years	24 (9.8)	222 (90.2)
Years, n (%)		
2017	47 (8.9)	478 (91.1)
2018	56 (10.8)	463 (89.2)
2019	167 (17.7)	777 (82.3)
2020	80 (15.0)	453 (85.0)
2021	85 (20.2)	335 (79.8)
Ethnicity, n (%)		
Malay	402 (15.4)	2204 (84.6)
Chinese	22 (12.6)	152 (87.4)
Indian	3 (6.8)	41 (93.2)
Other	8 (6.8)	109 (93.2)

**Table 2 ijerph-20-01848-t002:** Correlation of meteorological factors with RSV cases.

Meteorological Factors	Range	Mean ± SD	Correlation Coefficient ^a^	*p*-Value	b-Value ^b^	*p*-Value
Rainfall (mm)	72–678	308.5 ± 137.5	−0.05	0.007 **	−0.001	0.004 **
Rainy days (n)	9–29	17.9 ± 4.7	−0.047	0.010 **	−0.004	0.005 **
Temperature (°C)	26.7–29.4	28.4 ± 0.6	0.041	0.025 *	0.03	0.006 **
Relative humidity (%)	62.9–82.3	72.7 ± 3.9	−0.047	0.011 *	−0.005	0.006 **

^a^ Results of Spearman’s rank correlation. ^b^ Results of multiple bivariate logistic regression. * Significance at *p* < 0.05. ** Significance at *p* < 0.01.

## Data Availability

The data analyzed during the current study are available from the corresponding author upon reasonable request.

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
