# Peer review of "Determining the Relationship of Meteorological Factors and Severe Pediatric Respiratory Syncytial Virus (RSV) Infection in Central Peninsular Malaysia"

_ijerph, 2023, doi:10.3390/ijerph20031848_

Round 1

Reviewer 1 Report

The manuscript of Munchan and colleargues described the influence of rainy seasons onto the number of cases of RSV in Kuala Lumpur Malaysia. Although interesting, it cannot be published in International Journal of Environmental Research and public health without significant modifications.

Major comments:

+ Title : The authors should add ”Severe pediatric” before  RSV infections because only children requiring Non-invasive ventilation were included.

+ Methods : The restriction of activities due to SARS CoV2 pandemic has not been detailed before discussion. Were there lockdown in Malaysia? How many times and how long? Were restriction of activities limited to children or did the concern adults too? When did these restrictions of activity occur in 2020 and 2021?

+ Methods : what about duplicate inclusion of the same child several times over the study period ? Does this event occur? Was the second inclusion excluded?

+ Results : The results of Rainfall were consistent with those of rainy season or humidity . However, these variables all increased before the monsoon season reported in the introduction (march to june instead of May to August and October to February instead of November to February). Why? Is it due to global warming? This should be precised and further discussed.

+ Results : For my point of view, the two peak of RSV severe cases are July and August and December to March. These two peaks are in the end of the monsoon season, the first with higher temperature than the second one. This should be precised too.

+ Results : Why did the authors only use Spearmann rank correlation? It would have been useful to check results with Spearmann rank correlation with a logistic regression model and another categorical expression of the number of RSV cases, such as number of case>7/month yes-no

+ Results : The correlation and regression performed by the authors give a result for the whole years, whereas two disctinct peaks of RSV cases at the end of monsoon season with different temperature are observed. The results of this statistical analysis could be the sum of two different patterns. The authors should make supplementary separate analysis for the cases of yearly first peak and yearly second peak. If discordant results were shown, the impact of monthly activity of population (working period, school holidays, and religious or national celebrations period) or social habits (tobacco consumption) onto severe RSV pediatric infections should be discussed

+ Results : Because of restrictions of activity due to SARS CoV2 pandemics as well as mask wearing that could both have an impact onto the number of RSV infections,  the authors should provide supplementary analysis limited to 2017-2019 and to 2020-2021 time periods.

+ Discussion the authors should say earlier that their results conciliate previous results of the literature with two peaks of severe RSV infections and rewrite discussion taking into account supplementary results required above.

Minor comments:

+ Multiple syntax errors through the whole manuscript

e.g  abstract :

- more common is more commonly used than commonest; hotter, less humid and lowest is incorrect

e.g discussion :

- few, not a few line 194

+ abstract : prefer contradictory to scarce line 16

+ abstract : Palivizumab but also Nirsevimab

+ introduction line 46 “activation” please explain

+ discussion : limits of sensitivity of Direct Fluorescent antibody method has not been discussed

Reviewer 2 Report

Chan et al determined the relationship between RSV infection in Malaysia and meteorological factors such as rainfall, rainy days, temperature and relative humidity. They found that RSV infection peaked in two period when there was lowest rainfall and number of rainy days. They also found the positive association between RSV prevalence and temperature, and the negative association between RSV prevalence and humidity. The work is excellent as it supported important data to the administration for   RSV infection prevention and control.

 Major comments

1. The contents of “Materials and Methods” is ok. However, this chapter is not readable to readers. I recommend the authors to change the structure and add several subsections to organize the information. For example:

Study design and study population-How the study is designed?

RSV infection detection-Give a description of RSV detection. Is Respiratory Virus Screening and Identification Kit an ELISA assay?

Sample size-What the sample size look like from 2017 to 2021. The inclusive and exclusive criteria of samples.

Statistical analysis-Give some description of how the analysis was measured.

2. The chapter of result. I also recommend the authors separate the data into several subsection and make a summarized title of each subsection. For example:

Describe the rates of RSV infection in different age groups in the past 5-year period

The relationship between RSV infection and rainfall, number of rainfall days

The relationship between RSV infection and temperature, and humidity.

Minor comment

1. Table 2, line 173, I recommend use different numbers of asterisk to indicate the significant level <0.05, <0.01.

Round 2

Reviewer 1 Report

The authors have taken into account all my comments.

I would suggest three minor comments before publication:

-          Line 177-178 : 444 -8 = 435 cases?

-          The supplementary results should appear in the results part and not only in the discussion part. Why were the supplementary results about first and second monsoon not cited in the results part?

-          Taking into account epidemiological variability between places and countries especially at the time of global warming, maybe the authors should state in their discussion part that each centre/region should use previous local data about RSV peak incidences to better define the best period for Palivizumab (but also Nirsevimab) preventive campaigns

Author Response

Please see the attachment below
